# Advancing the Adoption of Continuing Professional Development (CPD) in the United States

**DOI:** 10.3390/pharmacy8030157

**Published:** 2020-08-31

**Authors:** James A. Owen, Jann B. Skelton, Lucinda L. Maine

**Affiliations:** 1American Pharmacists Association, Washington, DC 20037, USA; jowen@aphanet.org; 2American Association of Colleges of Pharmacy, Arlington, VA 22202, USA

**Keywords:** continuing professional development (CPD), technology, continuing professional education (CPE)

## Abstract

Over the last four decades, the expanded patient care roles of pharmacists in the United States (U.S.) have increased focus on ensuring the implementation of processes to enhance continuing professional development within the profession. The transition from a model of continuing pharmacy education (CPE) to a model of continuing professional development (CPD) is still evolving. As pharmacists assume more complex roles in patient care delivery, particularly in community-based settings, the need to demonstrate and maintain professional competence becomes more critical. In addition, long-held processes for post-graduate education and licensure must also continue to adapt to meet these changing needs. Members of the pharmacy profession in the U.S. must adopt the concept of CPD and implement processes to support the thoughtful completion of professional development plans. Comprehensive, state-of-the-art technology solutions are available to assist pharmacists with understanding, implementing and applying CPD to their professional lives.

## 1. Introduction

Pharmacists’ patient care roles continue to expand in the United States. To prepare for these enhanced roles, the pharmacy profession is building the infrastructure and developing processes to support workforce development, prepare pharmacists to meet new practice opportunities, and drive practice transformation. A primary focus is the need to shift from a system of typically undirected continuing professional education (CPE) to acceptance of, and engagement in, the holistic approach of continuing professional development (CPD). The pharmacy profession in the United States represents pharmacists in substantially diverse practice environments and patient care settings who operate with variation in scope and authority in different states across the country. Adoption of the principles of CPD is essential to enable pharmacists to develop, prepare, and maintain competency and proficiency in practice over time and to retool to support ongoing transformations in practice.

Within the profession in the United States, there is a gap in understanding and awareness of the differences between CPD and CPE, with most pharmacists believing that these concepts are interchangeable. The concept of CPD within the profession of pharmacy was initially defined by the International Pharmaceutical Federation (FIP) as “the responsibility of individual pharmacists for systematic maintenance, development and broadening of knowledge, skills and attitudes, to ensure continuing competence as a professional, throughout their careers” [1]. CPE is defined as: “a structured educational activity designed or intended to support the continuing development of pharmacists and/or pharmacy technicians to maintain and enhance their competence” [2].

While CPD for pharmacists has expanded globally, the universal adoption of the concept is still illusive. Over the past 3 years, professional pharmacy organizations in the United States have collaborated to develop unique and innovative technology solutions that collect, maintain, and verify pharmacists’ credentials and help pharmacists develop a comprehensive, yet individualized, professional development plan. The use of these interconnected technology systems will facilitate the adoption of CPD. Additionally, training student pharmacists about the tenents of CPD during their professional program will reinforce the value of ongoing professional growth and development and the value of establishing a professional identity.

## 2. The Evolution of the Practice of Pharmacy

Across the world, the practice of pharmacy is rapidly evolving. As part of this evolution, the United States has been transitioning from a product-centered distribution focus to a patient-centered care focus over the last four decades. There have been many factors that have contributed to this progress. A pivotal tipping point was the development and publication of the pharmaceutical care model by Hepler and Strand in 1990 [3]. The pharmaceutical care model facilitated the adoption of a patient-centered pharmacotherapy curriculum in many schools and colleges of pharmacy. Additionally, the adoption of the concept of pharmaceutical care by the Accreditation Council on Pharmacy Education (ACPE) set the standard expectation by which all accredited pharmacy schools and colleges would educate student pharmacists. This evolution of pharmacy education and the focus on the advanced role of the pharmacist ultimately transitioned the profession to an entry-level degree of the Doctor of Pharmacy (PharmD) in 2004 [4].

Over the last 20 years, since the transition to the entry-level PharmD, an increasing number of pharmacists are also pursuing post-graduate residency training [5]. Fellowship training programs are another avenue for post-graduate training intended to help candidates develop competency and expertise in the scientific research process. Approximately 37% of pharmacy school graduates plan to pursue post-graduate training; although there are not enough PGY1 residency programs or fellowship programs to meet this need [6]. In 2020, 5908 pharmacists applied for 3914 available PGY-1 residency positions [7]. Additionally, in the United States, there has been a steady increase in pharmacists pursing specialty certifications through the Board of Pharmacy Specialties, with approximately 10% of licensed pharmacists having one or more specialty certifications [8]. The increased interest in expanded pharmacist credentialing is correlated with the increased opportunities within patient care roles and reinforces the value of CPD to enhance the knowledge, skills, and abilities of an individual practitioner.

## 3. Expanded Patient Care Roles for Pharmacists in the United States

The advancement in the education, training, and certification of pharmacists has vastly expanded the scope, nature, and type of patient care services provided by pharmacists. As shown in Figure 1, the most prevalent services offered by pharmacists are medication therapy management, disease state education, immunizations, medication adherence services, disease state management services, care transition services, health and wellness screenings, and smoking cessation services [9].

As medication-related problems continue to be a substantial concern in achieving outcomes for patients, pharmacists are increasing their roles in the care and treatment of patients as members of the broader health care team. Groups outside of pharmacy, including the Centers for Medicare and Medicaid Services and the National Governors Association have recognized pharmacists as necessary providers in a transformed health care system where the quality, consistency, and outcomes for patients are improved [10,11]. Additionally, in an effort to expand medically underserved patients’ access to pharmacists’ services, the Patients Access to Pharmacists’ Care Coalition is working to develop and enact federal legislation that would enable patient access to, and reimbursement for, Medicare Part B services by state-licensed pharmacists in medically underserved communities consistent with state scope of practice law [12]. For pharmacists to be prepared to deliver enhanced pharmacist services for both current and emerging roles, processes to ensure the continued advancement of knowledge, skills, and abilities are required.

## 4. Expanded Definition of Community-Based Practice in the United States

Pharmacists in the U.S. work in highly diverse practice settings. In the 2019 Pharmacist Workforce Study conducted by the Pharmacy Workforce Center, the most predominant settings are community-based and hospital practices [13]. According to the U.S. Bureau of Labor Statistics, 57% of pharmacists work in community-based settings and 26% work in health-systems [14]. The diversity of community-based practice in the United States has been driven by the patient-centered focus of the U.S. health care system. Patients seek to obtain care most conveniently, when and where they want it. Community-based care settings serve as convenient, cost-effective, primary access points to care. The American Pharmacists Association (APhA) has published a model of community-based pharmacy practice that describes this practice as any patient care setting outside the four walls of a hospital or health-system (see Figure 2). The term “community-based pharmacist practitioner” highlights the unique skillset that pharmacists use when they provide patient care services, and it brings recognition to the value community pharmacists contribute to patients, communities, and the health care system [15,16].

## 5. Regulation of Pharmacy Practice in the United States

The complexity of pharmacy practice within the United States is driven primarily by the regulatory process. While there are some commonalities, pharmacy practice is governed by the laws, regulations, and rules set forth by each individual state or jurisdiction. What is permitted in one jurisdiction may be prohibited by another, and even states geographically close may have substantially different authorities in scope of practice. An example of this variability is in the laws and regulations that empower pharmacists to immunize. In some states, pharmacists can only immunize according to an established protocol, in other states by a protocol and a prescription, and still other states with no authorization, depending upon the type of vaccination and age of the patient. Additionally, the restrictions on immunizations provided by pharmacists also extend to permitting immunizations of patients of various ages, ranging from greater than three years to 18 years old [17].

The variability of practice laws, regulations, and rules expand across many areas of practice. Another emerging area of practice in the United States is the initiation of therapy via prescribing under statewide protocols, statewide standing orders, or unrestricted authority. As demonstrated in Figure 3, data collected by the National Alliance of State Pharmacy Associations, there is variability in practice scope across the country, with the broadest prescriptive authority provided in the State of Idaho with authorization for pharmacists to prescribe medications in greater than 20 categories [18].

As the breadth and depth of pharmacist services increase and the scope of practice and authority for pharmacists continues to rapidly expand, there is a pressing need for pharmacists to engage in intentional, purposeful, and directed professional development focused on the specific needs of the individual pharmacist.

## 6. Assuring the Professional Competency of Pharmacists

Responsibility for the quality of pharmacy education in the United States is overseen by ACPE, which was initially founded in 1932 in collaboration with APhA, the American Association of Colleges of Pharmacy (AACP), the National Association of Boards of Pharmacy (NABP), and the American Council on Education. ACPE serves as the autonomous body for the accreditation of schools and colleges of pharmacy and providers of continuing education and is the only accreditation program recognized by the U.S. Department of Education to accredit pharmacy professional degree programs.

NABP oversees U.S. pharmacist license requirements and administers the North American Pharmacist Licensure Examination (NAPLEX^®^) and the Multistate Pharmacy Jurisprudence Examination (MPJE^®^). NABP facilitates the interstate/inter-jurisdiction transfer or reciprocation of a pharmacist’s licensure, based upon a minimum standard for pharmacy education and uniform legislation requirements. State boards of pharmacy manage and maintain records for licensure of individual pharmacists.

The established and predominant model for pharmacist re-licensure is built upon the concept of CPE, which was developed to align with continuing education requirements for other health care professionals. CPE for pharmacy was initially discussed in the United States in the 1940s, and in 1965, Florida became the first state to mandate CPE for pharmacist licensees [19]. From 1972 to 1974, the APhA and AACP Task Force on Continuing Competency in Pharmacy concluded that “CPE was the best mechanism to assure pharmacist proficiency” [20]. In 1974, NABP adopted a formal resolution on mandatory continuing education for re-licensure. That same year, the APhA Board of Trustees recommended ACPE develop a system for accreditation of CPE providers. In 1975, ACPE introduced the initial standards for CPE. Beginning in 1975, all states and territories in the United States now require completion of a specified amount of ACPE-accredited or state-approved CPE for license renewal [2]. As with state practice laws, regulations, and rules, requirements for CPE are also variable by individual state or jurisdiction.

Historically, reporting, tracking, and maintenance of CPE records were considered the responsibility of individual pharmacists. Each state has requirements for submitting documentation, reporting CPE completion, or attesting to meeting the requirements established for re-licensure. In 2011, NABP and ACPE partnered on CPE Monitor^®^, a technology solution to help pharmacists and state boards of pharmacy track and manage the completion of CPE required for state re-licensure. The CPE Monitor^®^ system changed the reporting requirements from pharmacists to the CPE providers and required all pharmacist users to register with and obtain an e-Profile with NABP. This service enabled state boards of pharmacy to access license completion data for ACPE-accredited CPE programs as a means of tracking compliance with CPE completion by licensees in the state’s jurisdiction [21].

While CPE is beneficial in contributing to the maintenance of professional competency, questions remain about the benefits of CPE alone in assuring clinical competence in practice [22]. Globally, the methods used to assure sustained professional competencies for pharmacists are diverse and variable. Some countries, such as Canada and Japan, ensure competence through on-going and continued assessments tied to retaining or expanding the authority to practice. Other countries, such as the United Kingdom, Ireland, and others, have adopted models of CPD with varying degrees of success [23]. The United States has been slow to adopt CPD, with only the states of Iowa and North Carolina recognizing CPD, in addition to CPE, as part of their licensure requirements. All other states only recognize CPE within their licensure criteria.

## 7. Health Care Practitioner Responsibilities for Ensuring Continuing Professional Competency

In the profession of medicine, the American Medical Association (AMA) describes the responsibility for ensuring continuing professional competency in the AMA Code of Medical Ethics. The AMA Code of Medical Ethics states, “a physician shall continue to study, apply, and advance scientific knowledge, maintain a commitment to medical education, make relevant information available to patients, colleagues, and the public, obtain consultation and use the talents of other health care professionals when indicated” [24].

In pharmacy, APhA states in the Code of Ethics for Pharmacists, adopted in 1994, that, “A pharmacist maintains professional competence. A pharmacist has a duty to maintain knowledge and abilities as new medications, devices, and technologies become available, and as health information advances” [25]. Globally, the FIP Code of Ethics for Pharmacists, adopted in 1997, states pharmacists have a responsibility “to ensure competency in each pharmaceutical service provided by continually updating knowledge and skills. [26] Additionally, the codes of ethics of other health care practitioners, including nurses and physician assistants, share the same perspective of protecting the care and welfare of patients in the provision of patient care by ensuring on-going, continuing professional competency maintenance and development throughout the health care practitioner’s career [27,28].

### Transitioning from a Continuing Education Model to a Continuing Professional Development Model in the United States

In 2010, the Institute of Medicine (IOM) published the report Redesigning Continuing Education in the Health Professions. This report stated, “all health professions should move toward requiring licensed health professionals to demonstrate periodically their ability to deliver patient care through direct measures of technical competence, patient assessment, evaluation of patient outcomes, and other evidence-based assessment methods”. The IOM report published the following statements [29]:▪A workforce of knowledgeable health professionals is critical to the discovery and application of health care practices to prevent diseases and promote well-being▪The absence of a comprehensive and well-integrated system of continuing education in the health professions is an important factor of knowledge and performance deficiencies at the individual and system levels▪The new vision for continuing education will be based on an approach called CPD in which learning takes place over a lifetime and stretches beyond the classroom to the point of care▪The CPD system is a holistic approach that incorporates a broader variety of learning methods and theories than CE▪CPD is learner-driven and tailored to the individual learner’s needs; it includes concepts such as self-directed learning and practice-based learning and teaches how to identify problems and apply solutions

While CPD existed in the health professions before the IOM report in 2010, over the last decade, increased focus on the implementation of these recommendations have assisted many health professions to refocus efforts on implementing CPD to ensure the clinical competence of members in their respective disciplines.

## 8. Advancing Continuing Professional Development in the Pharmacy Profession

One of the earliest advancements of CPD in the pharmacy profession can be traced to work completed by FIP in 2002. The FIP Statement of Professional Standards, Continuing Professional Development, stated, “pharmacists are health care professionals whose professional responsibilities include seeking to ensure that people derive maximum therapeutic benefit from their treatments with medicines. This requires them to keep abreast of developments in pharmacy practice and the pharmaceutical sciences, professional standards requirements, the laws governing pharmacy and medicines and advances in knowledge and technology relating to the use of medicines. This can only be achieved by an individual’s personal commitment to Continuing Professional Development.” The FIP model for CPD is depicted in Figure 4. The FIP model defines the following requirements for CPD [30]:▪The CPD process is defined as “the responsibility of individual pharmacists for systematic maintenance, development and broadening of knowledge, skills and attitudes, to ensure continuing competence as a professional, throughout their careers”▪CE is an important part of a structured CPD program, personalized for each pharmacist▪The CPD process should be visible to ensure credibility with the public▪CPD must be actively managed and include all the components of the cyclical process▪CPD must be an on-going, cyclical process of continuing quality improvement by which pharmacists seek to maintain and enhance their competence in both current duties and anticipated future service developments

In the United States, the Council on Credentialing in Pharmacy (CCP), the body that serves to coordinate credentialing and CPD for the profession, expanded on the work published by FIP and in 2004 released The Council on Credentialing in Pharmacy Resource Document, Continuing Professional Development in Pharmacy. This document was aligned with the model published by FIP two years earlier. CCP summarized that the need for CPD is to [31]:▪Ensure that pharmacists maintain (at an appropriate level) their knowledge, skills, and competence to practice through their careers in their own specific (or current) areas of practice▪Improve the pharmacist’s personal performance (i.e., develop knowledge and skills)▪Enhance the pharmacist’s career progression

While aligned with the FIP model, the CCP model was modified slightly to meet the identified needs of pharmacists in the United States. The CCP Model is shown in Figure 5.

In January 2015, ACPE published the Accreditation Council for Pharmacy Education Guidance on Continuing Professional Development (CPD) for the Profession of Pharmacy. This guidance provided explanation and clarification of the differences between CPE and CPD and offered further refinement of the model published by CCP in 2004. Importantly, the ACPE Guidance provided specific categories and examples of CPD activities that helped to better explain to pharmacists and the pharmacy profession the types and scope of activities associated with CPD. The model published in the ACPE Guidance is shown in Figure 6 [32].

## 9. The Current U.S. Practice Environment

Pharmacy in the United States is constantly changing, and the job market is extremely competitive. Pharmacists must engage in lifelong learning to stay relevant, meet licensure requirements and complete activities that employers assign. Pharmacists also have individual professional goals that may support seeking a promotion or changing career paths.

One of the most challenging concepts for pharmacists to adopt is that CPD is broader than participation in CPE. As stated in an article published in *JAMA* in 1999, “CE on its own, does not necessarily lead to positive changes in professional practice, nor does it necessarily improve health outcomes” [33]. Despite almost twenty years since the work conducted by FIP in pharmacy, and the publication of resources and guidance by both CCP and ACPE, there remains little awareness or understanding in the United States among practicing pharmacists about the differences between CPE and CPD.

## 10. Challenges with the Adoption of CPD

A 2014 study conducted by FIP gathered data and reported detailed findings from 66 countries about their experiences with CPD. This report documented the following [34]:▪“The CPD approach to learning is embraced more by those with a mentor or colleague demonstrating use or those that incorporated a similar approach during their undergraduate education”▪“The flexibility of the CPD approach allows those with specialized practice to engage in learning that is most beneficial to their particular practice (e.g., non-pharmacy conferences, workplace learning, research, etc.)”▪“There is value in focusing on an outcome in practice (i.e., applying newly-acquired knowledge to practice)”▪“Pharmacists support the CPD process when the impact in practice is recognized”▪“The CPD cycle enables a continuous learning process and becomes a valuable platform to reinforce continuing competence within practice”

The report also outlined the challenges that different countries experienced with the successful implementation of CPD. Examples of these challenges detailed by FIP include [34]:▪“Application of the CPD cycle as an approach to lifelong learning in individual practice”▪“Changing the focus of CPD to meaningful learning rather than collecting CE points”▪“Developing the skills and knowledge to adopt the CPD process under a voluntary system”▪“Establishing adequate assessment methods and measures for a portfolio-based system”▪“Raising the awareness and recognition from the public on the skills and knowledge that pharmacists continuously develop”▪“Ensuring the qualitative assessment of CPD portfolios”▪“Lack of technical ‘know-how’ for implementing a robust online system to support the CPD process”▪“Ensuring pharmacists engage with the CPD cycle that includes reflection on their practice”▪“Minimizing cost constraints for pharmacists to undertake formal CPD programs”▪“Making the recording or documentation of learning less onerous”▪“Identifying those pharmacists who require additional mentoring or guidance to steer away from the hour or point collecting mentality”▪“Ensuring relevant support for pharmacists in non-traditional roles”

This report collectively demonstrated that pharmacists engaged in CPD had improved the quality of their learning, which leads to improved self-assessment of learning needs and enhancements in overall pharmacy practice. The report identifies challenges to the widespread implementation of CPD. Pharmacists are often unfamiliar with the CPD process and the time commitment required for implementation. Pharmacists trained on CPD and who used tools to facilitate the process were more likely to be successful. The availability of tools and resources that support pharmacist engagement in CPD is imperative to enable expanded implementation. According to the report, competency standards, pharmacist learning plans, and competency mapping of CPD activities must also be more effectively linked. Finally, efforts to raise awareness of the differences between CE and CPD are required [35]. Adoption of technology solutions can help provide support to pharmacists wishing to engage in CPD and minimize the challenges associated with CPD adoption.

## 11. Technology Solutions to Support the Adoption of CPD

Specifically in the United States, a different approach toward implementation of CPD may be required of pharmacists, CE providers, employers/institutions, and regulators. These entities should recognize that many pharmacists will need to gain new expertise, such as identifying individual learning needs, writing SMART learning objectives, and developing personal learning plans to implement CPD in practice [35,36]. Infrastructure is also needed to help support workforce development and ensure that the profession is equipped to rapidly assess pharmacist readiness and retool pharmacist learning. This issue has recently been highlighted in the United States, with all organizations and individual pharmacists engaging in professional development activities to learn to care for patients with COVID-19.

To help pharmacists better understand and embrace the tenants of CPD, several professional pharmacy organizations have developed technology solutions to support pharmacist adoption of CPD.

## 12. My CPD^®^

Over the last five years, ACPE has undertaken substantial work to communicate the value of CPD in the United States. Through ACPE Board of Directors’ directed projects, task forces and initiatives, and a partnership to develop an app called CPE Monitor Plus^®^, ACPE launched an initial technology solution called My CPD^®^ that permits users to create and manage a CPD plan. CPE Monitor Plus^®^ was launched in April 2018 and is available as a subscription service through purchase of the app by the individual pharmacist [37].

My CPD^®^ is a web-based platform that allows pharmacists to create, record and maintain CPD details. Pharmacists build and evaluate personal development plans, document learning activities, and upload supporting evidence of learning and its impact in practice. My CPD^®^ tracks both accredited CE as well other CPD activities undertaken in order to maintain and advance competencies in areas relevant to professional responsibilities. Currently, the My CPD^®^ platform does not support personal assessments but encourages the development of individual goals and objectives.

The My CPD^®^ app provides an overview of professional development and helps pharmacists create CPD cycles and review professional progress over time. According to the My CPD^®^ website, MyCPD^®^ allows pharmacists to:▪Ensure that knowledge, skills, attitudes, and values are kept current▪Apply learning to improve practice and/or patient outcomes▪Direct career goals or support career advancements or changes▪Provide CV or interview templates▪Fulfill employment, regulatory and credentialing requirements

## 13. ADVANCE^®^

In late 2015, APhA began developing a state-of-the-art web-based technology solution, ADVANCE^®^, to enable pharmacists to proactively engage in the CPD process. Designed by and for pharmacists, ADVANCE^®^ was developed with the end-user in mind, taking a complex process and simplifying it using interactive and intuitive technology solutions. ADVANCE**^®^** is an online professional development planner for pharmacists to easily create a personalized plan. ADVANCE**^®^** takes the complex process of professional development planning and maintenance and simplifies and streamlines the process [38]. The ADVANCE**^®^** process is aligned with the FIP, CCP, and ACPE models and provides a pharmacist-friendly user interface that supports pharmacists through the process to complete their annual professional development plan.

As shown in Figure 7, all users complete five steps to develop a CPD plan. These steps include:▪Evaluate—Guides pharmacists through seven evaluation exercises to determine professional development needs▪Analyze—Helps pharmacists determine the meaning of the information provided in the Evaluate step and provides a framework to establish SMART planning goals▪Plan—Supports pharmacists by recommending activities and experiences customized to the individual, from which they can create a comprehensive professional development plan▪Do—Provides a framework to help pharmacists set priorities and deadlines for all selected activities, track accomplishments, and maintain a professional portfolio▪Apply—Prompts pharmacists to reflect on the activities they have completed, consider what they may want to focus on next, and make updates to the professional development plan

The identification of personal learning needs is a core feature of this CPD platform. Pharmacists are guided through assessments that help users honestly and comprehensively assess their current status and determine professional development needs. These assessments include:▪Strengths—Leveraging personal strengths can help pharmacists better understand themselves, how they work, and how they contribute to a team.▪Professional Skills—Each individual has a unique set of professional skills shaped by past experiences, interests, and goals. Like many aspects of life, professional skills may change or grow over time. This assessment leverages the Entrustable Professional Activities and allows pharmacists to assess both their baseline and progress toward improving their skills.▪Personal Attributes—Personal attributes are associated with how an individual’s personality, emotions, and tendencies contribute to their behaviors. These attributes can significantly impact a professional’s ability to interact with others and function effectively.▪Interest Areas—Aligning an individual’s career with their interests increases the likelihood that they will achieve professional satisfaction and maximizes the impact of CPD.▪Growth and Development—The technology platform evaluates each user’s assessments and provides customized areas for growth and development focused on the improvement of professional skills and personal attributes.▪Licensure—Evaluating a summary of your licensure requirements supports pharmacists in designing a professional development plan that simultaneously keeps professional licenses active and individuals moving toward their goals.▪Well-Being—Pharmacist well-being is essential to personal and professional productivity, and understanding factors that create stress or support relaxation can positively impact both the personal and professional needs of pharmacists.

Pharmacists are guided to review and analyze what they have discovered through these assessments, interpret the information, and examine each area’s importance to their personal and professional development. This assessment process allows for the development of a more customized and valuable CPD plan and supports the prioritization of established goals.

The CPD plan is comprised of six types of activities and experiences, which include patient care, management, leadership, community service, scholarly activity and well-being. By harmonizing skills, interests, strengths, and attributes with licensure and employer requirements, ADVANCE**^®^** recommends specific activities and experiences that are aligned to an individual pharmacist and helps make CPD more attainable for practicing pharmacists. Recommended activities might include CPE programs, books, videos, articles, community events, advocacy, or a variety of other actions.

The technology allows pharmacists to prioritize and track activities and reflect on experiences and accomplishments. Maintenance of a personal portfolio is stressed, with tools that allow pharmacists to attach relevant information and documents to their CPD plan. Users are prompted to keep their plan current by reviewing information, documenting progress and making updates to the plan at least quarterly. Simply stated, ADVANCE**^®^** takes the complicated CPD process, which is foreign to most pharmacists, and makes CPD accessible, intentional, understandable and usable for practicing pharmacists.

## 14. ADVANCE^®^ Recognition

In the Report of the 2019–2020 AACP Academic Affairs Committee, the organization outlined its plans to align with other organizations and associations providing CPD initiatives focused on the large-scale pharmacist workforce. AACP also recently proposed policy to support ADVANCE**^®^** as a novel approach to enabling workforce development [39].

## 15. CPD Model Comparison

Although available models of CPD may vary slightly, the published processes are similar and all provide support to a pharmacist engaging in CPD. Table 1 provides a crosswalk to the models, the steps, and the functions of the CPD models.

## 16. Supporting the Advancement of CPD in the United States

As pharmacists throughout the United States expand the nature and complexity of patient care services and push for enhanced authority within their jurisdictions, the need to maintain and demonstrate competency and proficiency remains critical. While CPD alone will not lead to improved care and outcomes, it will promote the overall growth and development of professional knowledge, skills, and abilities leading to enhanced professional competency. Organizations focused on encouraging and supporting pharmacist engagement in CPD should reinforce the importance of keeping knowledge, skills and abilities current. For many pharmacists, the concept of CPD is novel, and pharmacists should be encouraged to use any process or program they find effective in supporting and advancing their CPD.

Lifelong learning encompasses all learning activities undertaken throughout life, with the aim of improving knowledge, skills and competencies within a personal, civic, social and/or employment related perspective [40]. Research done during the development process for ADVANCE^®^ reinforced the value of supporting pharmacist assessments of personal character attributes, which include skills such as leadership, management, accountability, conflict resolution and collaboration. Personal character attributes contribute to a pharmacist’s professional identity and are also an important component of professional development. The re-professionalization of pharmacists in the United States is needed, and adoption of CPD models with supporting technology can facilitate awareness, understanding, and uptake by pharmacists.

## 17. Conclusions

As the roles of pharmacists continue to expand in the United States and globally, pharmacists must recommit to their personal and professional development. Members of the pharmacy profession must adopt the concept of CPD and implement processes to support the thoughtful completion of professional development plans. Comprehensive, state-of-the-art technology solutions are available to assist pharmacists with understanding and implementing CPD in their professional lives.

Additionally, systems such as ADVANCE^®^ provide the profession with a comprehensive solution to track the progress of individuals and populations over time, which may assist the profession in better meeting the development needs of the profession as a whole. To facilitate the adoption and acceptance of CPD in the United States, it is recommended that technolgies such as CPE Monitor Plus^®^ and ADVANCE^®^ be adopted to simplify the task of engaging in CPD by both practicing pharmacists and student pharmacists in the United States.

## Figures and Tables

**Figure 1 pharmacy-08-00157-f001:**
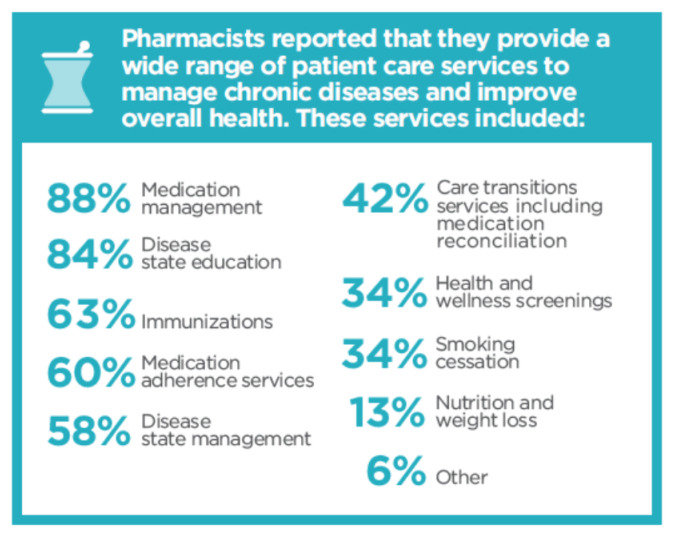
Patient Care Services Provided by Pharmacists in the United States Reprinted with permission from the American Pharmacists Association.

**Figure 2 pharmacy-08-00157-f002:**
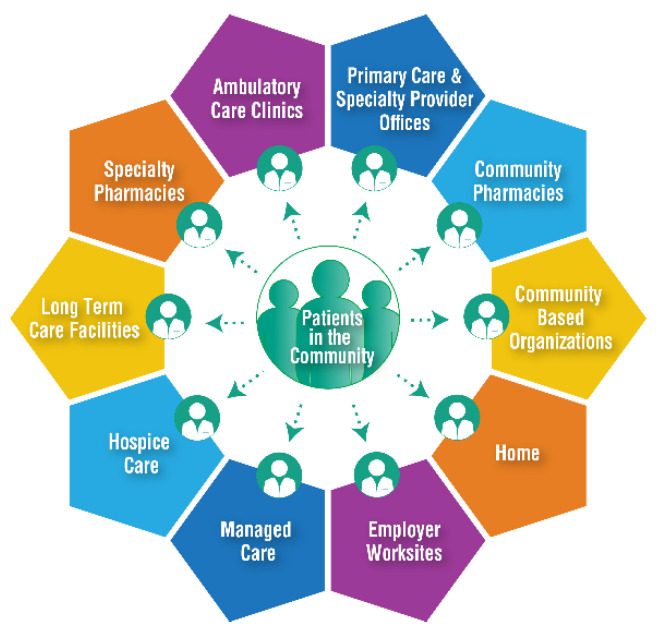
Reach of community-based pharmacist practitioners. Reprinted with permission from the American Pharmacists Association.

**Figure 3 pharmacy-08-00157-f003:**
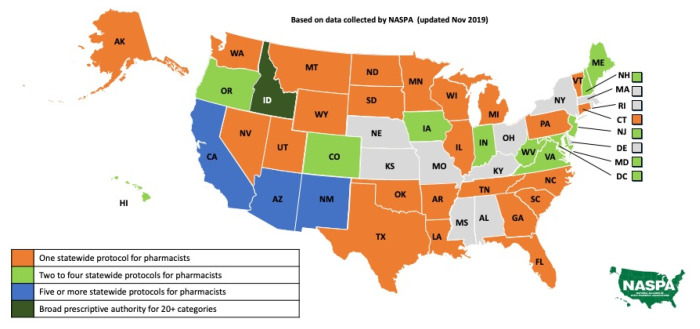
Prescribing under a statewide protocol, statewide standing order or unrestricted (category-specific) authority. Reprinted with permission from the National Alliance of State Pharmacy Associations.

**Figure 4 pharmacy-08-00157-f004:**
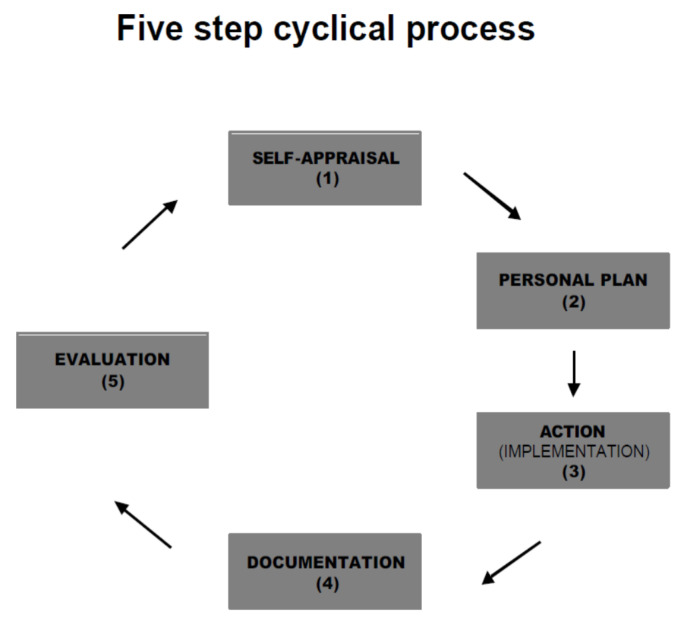
FIP continuing professional development process. Reprinted with permission from the International Pharmaceutical Federation.

**Figure 5 pharmacy-08-00157-f005:**
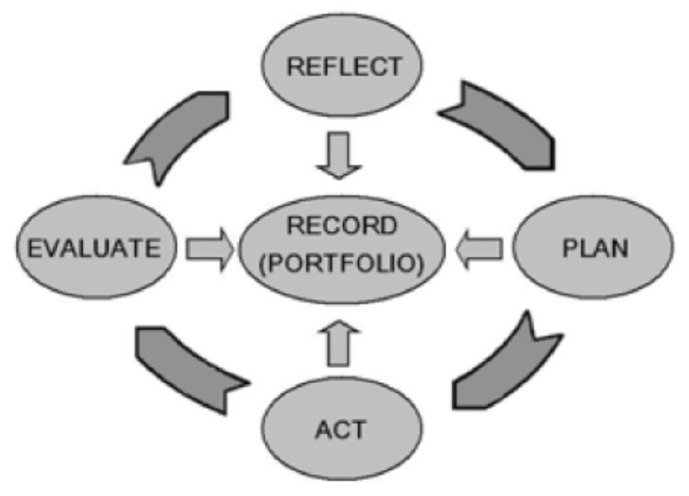
Council on Credentialing CPD model. Reprinted with permission from the Council on Credentialing.

**Figure 6 pharmacy-08-00157-f006:**
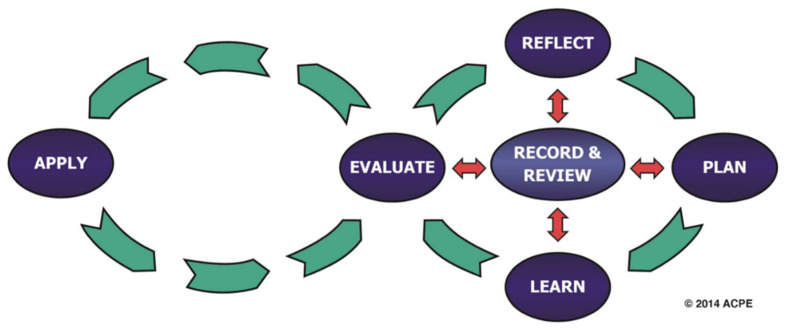
ACPE’s CPD cycle. Copyright © 2020–2014 Accreditation Council for Pharmacy Education. Used with permission.

**Figure 7 pharmacy-08-00157-f007:**
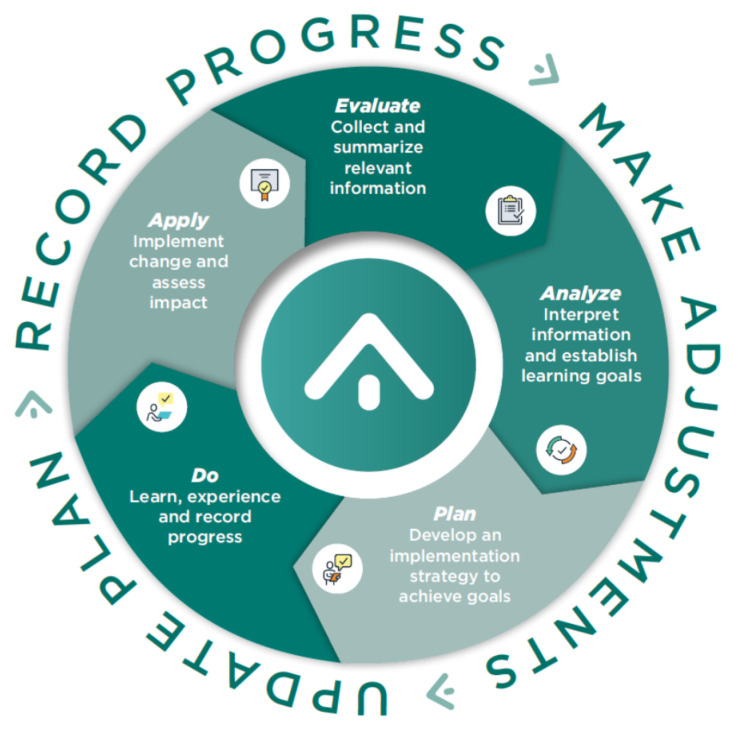
ADVANCE**^®^** CPD process. Reprinted with permission from the American Pharmacists Association.

**Table 1 pharmacy-08-00157-t001:** CPD model crosswalk.

FIP MODEL	ACPE MODEL	ADVANCE^®^ MODEL	FUNCTION
**Self-Appraisal**	**Reflect**	**Evaluate Analyze**	Completion of an annual self-evaluation—collect and analyze/assess relevant information and determine meaning
**Personal Plan**	**Plan**	**Plan**	Creation of a detailed action plan specifying activities and experiences associated with each area of CPD
**Action Implement**	**Learn**	**Do**	Engage in activities and experiences and track progress
**Evaluation**	**Evaluate Apply**	**Apply**	Implement what was learned and reflect on the impact. Routinely review and update information and make necessary modifications to plan.

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
