# Peer review of "Advancing the Adoption of Continuing Professional Development (CPD) in the United States"

_pharmacy, 2020, doi:10.3390/pharmacy8030157_

Round 1
Reviewer 1 Report
This is an interesting paper bringing together the historical and current situation for CPE/CPD within the USA. The structure and presentation of the work is to a high standard.
I would like to have had some more information about the app and how this works in terms of collecting together individual pharmacist's data, and I also would like to know what/how the initial personal assessments are undertaken.
In the introduction, there is a lack of reference to relevant literature in the early part of the section.
I think that the paper brings together a lot of information about CPD and makes the case for the move from CPE to CPD.
In terms of amendments, I think that the figures all need to have references added to them where relevant, as they all seem to be from other sources and I think this needs to be overtly recognised.
There are a few more detailed issues given below:
Line 88 - needs to be "are required" and not "is required"
Line 126 - needs to state "to engage in"
Line 140 & 158 - does Boards of Pharmacy need capitals as it is a title of a professional body
Line 168 - needs referencing to countries that need regular examination for ongoing registration
Line 188 - this statement needs a reference
Lin 189-190 should this be in bold as it is a heading?
Line 214 - FIP - title should be used in full the first time
Line 245 - should say figure 5 and not 4
Line 258, 259 & 406 - remove the comma before and
line 310 - reference the report referred to here
line 349 - comma needed after mind
Author Response
Responses to Reviewer 1 Comments
The authors wish to thank the reviewer for their time and expertise in reviewing the manuscript. Below please find a summary of the adjustments made to the manuscript in response to this feedback.
- Comment: I would like to have had some more information about the app and how this works in terms of collecting together individual pharmacist's data, and I also would like to know what/how the initial personal assessments are undertaken.
- Resolution: Added explanatory text “Currently, the MyCPD platform does not support personal assessments.” [line 396-397]
- Comment: In the introduction, there is a lack of reference to relevant literature in the early part of the section.
- Resolution – Additional detail in the form of definitions have been added to support the commentary provided by the authors in the introduction [lines 38-43]
- Comment: In terms of amendments, I think that the figures all need to have references added to them where relevant, as they all seem to be from other sources, and I think this needs to be overtly recognised.
- Resolution – Authors have obtained permission for reprint for each figure contained in the manuscript
- Figure 1: James A. Owen, VP, Practice and Science Affairs, American Pharmacists Association
- Figure 2: James A. Owen, VP, Practice and Science Affairs, American Pharmacists Association
- Figure 3: Rebecca Snead, Executive Vice President, National Alliance of State Pharmacy Associations
- Figure 4: Lin‑Nam Wang, Head of Corporate Communications & Advocacy, International Pharmaceutical Federation (FIP)
- Figure 5: Sam Johnson, President, Council on Credentialing
- Figure 6: Mike Rouse, Assistant Executive Director, Accreditation Council for Pharmacy Education
- Figure 7: James A. Owen, VP, Practice and Science Affairs, American Pharmacists Association
- Comment: Line 88 - needs to be "are required" and not "is required"
- Resolution: Revised as suggested
- Comment: Line 126 - needs to state "to engage in"
- Resolution: Revised as suggested
- Comment: Line 140 & 158 - does Boards of Pharmacy need capitals as it is a title of a professional body
- Resolution: Boards of pharmacy, when referring to globally to the concept should not be capitalizing. Referencing specific Boards would require capitalization e.g., Alabama Board of Pharmacy
- Comment: Line 168 - needs referencing to countries that need regular examination for ongoing registration
- Resolution: Added additional context in lines 197-198 that support the concept of ongoing assessments to retain or expand authority
- Comment: Line 188 - this statement needs a reference
- Resolution: Additional content added to include nursing and PAs; appropriate references included;
- Comment: Line 189-190 should this be in bold as it is a heading?
- Resolution: Revised as suggested
- Comment: Line 214 - FIP - title should be used in full the first time
- Resolution: FIP is spelled out at first use (Line 183)
- Comment: Line 245 - should say figure 5 and not 4
- Resolution: Corrected as suggested
- Comment: Line 258, 259 & 406 - remove the comma before and
- Resolution: Revised as suggested
- Comment: line 310 - reference the report referred to here
- Resolution: Requested reference added
- Comment: line 349 - comma needed after mind
- Resolution: Requested reference added
- Resolution – Authors have obtained permission for reprint for each figure contained in the manuscript
Reviewer 2 Report
The authors gave a thorough description of the differences between CPD and CPE in the US and the potential implications of the gaps in understanding for American Pharmacists. I particularly enjoyed exploring the author's argument that the increase in pharmacist's pursing specialty certifications through BPS, etc reinforces the value of CPD. One question that came to mind is the value of credentialing (which was mentioned in the paper). While I agree that this reinforces the value of CPD, I also think that the role of exclusivity of some of these educational certifications may also play a role in their demand (especially in new grads looking for good jobs). At least this is what I see locally.
I appreciated the descriptions of the AMA and APhA Code of Ethics. I wondered how other professions are dealing with the differences between CPD and CPE as I suspect this is an issue for them as well.
minor point: line 429 "soft skills" - perhaps the authors could find a better term for these important skills as I think the term soft is outdated and minimizes their importance
Author Response
The authors wish to thank the reviewer for their time and expertise in reviewing the manuscript. Below please find a summary of the adjustments made to the manuscript in response to this feedback.
- Comment: The authors gave a thorough description of the differences between CPD and CPE in the US and the potential implications of the gaps in understanding for American Pharmacists. I particularly enjoyed exploring the author's argument that the increase in pharmacist's pursing specialty certifications through BPS, etc reinforces the value of CPD. One question that came to mind is the value of credentialing (which was mentioned in the paper). While I agree that this reinforces the value of CPD, I also think that the role of exclusivity of some of these educational certifications may also play a role in their demand (especially in new grads looking for good jobs). At least this is what I see locally.
- Resolution: We very much appreciate the comment and perspective of the reviewer. We do not believe this commentary was intended to support a specific revision within the manuscript.
- Comment: I appreciated the descriptions of the AMA and APhA Code of Ethics. I wondered how other professions are dealing with the differences between CPD and CPE as I suspect this is an issue for them as well.
- Resolution: Included text that also highlights the focus of nurses and physician assistants toward CPD in their code of ethics. [line 226]
- Comment: line 429 "soft skills" - perhaps the authors could find a better term for these important skills as I think the term soft is outdated and minimizes their importance
- Resolution: Revised to read “Research done during the development process for ADVANCE reinforced the value of supporting pharmacist assessments of personal character attributes which include skills such as leadership, management, accountability, conflict resolution and collaboration. Personal character attributes contribute to a pharmacist's professional identity and are also an important component of professional development.” [lines 505-506]
Reviewer 3 Report
See attached document.

Author Response
The authors wish to thank the reviewer for their time and expertise in reviewing the manuscript. Below please find a summary of the adjustments made to the manuscript in response to this feedback.
- Comment: Consider incorporating a definition for CE and CPD in the introduction
- Resolution: Additional detail in the form of definitions have been added to support the commentary provided by the authors in the introduction [lines 38-43]
- Comment: Line 61-63 – The authors report that there are not enough PGY1 residency programs to meet the needs. This is certainly true in other countries. If the data is readily available, I would suggest the authors consider reporting the proportion of individuals who apply for PGY1 residency programs and are successfully matched.
- Resolution: Included current statistics - “In 2020, 5,908 pharmacists applied for 3,914 available PGY-1 residency positions.” [line 72-73]
- Comment: Line 90-93: If the data is available, consider presenting the % of pharmacists in each of the top 2 most common settings (community and hospital)
- Resolution: Included current statistics – “According to the U.S. Bureau of Labor Statistics, 57% of pharmacists work in community-based settings and 26% work in health-systems.” [lines 109-110]
- Comment: Line 212-256: Advancing CPD in the Pharmacy Profession - This section is dense and very detailed. Suggest considering whether each of the detailed bullet point definitions are required. Alternatively, the authors could consider condensing this information into a more easily digested comparison table or figure depicting a timeline of the relevant documents to allow the reader to visually see a progression. The authors could consider collapsing figures 4-6 into a single figure with side by side processes to allow for easier comparison.
- Response: The authors have thoughtfully included detailed information from published reports, as we believe that highlighting the value and importance of the original publications and will help direct inform readers who are exploring CPD as a concept and encourage access these sentinel publications. We have obtained written permission from each organization to reprint each figure. We are concerned that recasting or altering these figures may not be supported by the originators.
- Comment: Line 245: I believe this should read “The CCP model is shown in Figure 5”
- Resolution: Corrected as suggested
- Comment: Line 311: Technology Solutions to Support the Adoption of CPD. Suggest including a summary of differences or advantages/disadvantages of MYCPD vs. ADVANCE
- Resolution: The authors have added clarifying text in this section to differentiate the two available products without creating the perception of “competition” between these important tools to support pharmacists in their CPD.
- Comment: Line 420: Supporting advancement of CPD in the US; The authors could consider providing suggestions/considerations for those engaged in the development of CPD training programs in either this section or the conclusion.
- Resolution: The following clarifying text was added – “organizations focused on encouraging and supporting pharmacist engagement in CPD should reinforce the importance of keeping knowledge, skills and abilities current. For many pharmacists, the concept of CPD is novel, and pharmacists should be encouraged to use any process or program they find effective in supporting and advancing their CPD. [Line 497-501]